# The Presence of Polysaccharides, Glycerol, and Polyethyleneimine in Hydrogel Enhances the Performance of the Glucose Biosensor

**DOI:** 10.3390/bios9030095

**Published:** 2019-07-30

**Authors:** Marco Fois, Paola Arrigo, Andrea Bacciu, Patrizia Monti, Salvatore Marceddu, Gaia Rocchitta, Pier Andrea Serra

**Affiliations:** 1Dipartimento di Scienze Mediche, Chirurgiche e Sperimentali, Università degli Studi di Sassari, Viale San Pietro 43/b, 07100 Sassari, Italy; 2Dipartimento di Agraria and Unità di Ricerca Istituto Nazionale di Biostrutture e Biosistemi, Università degli Studi di Sassari, Viale Italia 39, 07100 Sassari, Italy; 3Istituto CNR di Scienze delle Produzioni Alimentari—UOS Sassari, Traversa La Crucca 3, 07100 Sassari, Italy

**Keywords:** glucose biosensor, glycerol, starch-based hydrogel, polyethyleneimine

## Abstract

The use of amperometric biosensors has attracted particular attention in recent years, both from researchers and from companies, as they have proven to be low-cost, reliable, and very sensitive devices, with a wide range of uses in different matrices. The continuous development of amperometric biosensors, since their use involves an enzyme, is specifically aimed at keeping and increasing the catalytic properties of the loaded protein, so as to be able to use the same device over time. The present study aimed to investigate the impact of glycerol and polysaccharides, in the presence of polycationic substances to constitute a hydrogel, in enhancing the enzymatic and analytic performance of a glucose biosensor. Initially, it was possible to verify how the deposition of the starch-based hydrogel, in addition to allowing the electropolymerization of the poly(p-phenylenediamine) polymer and the maintenance of its ability to shield the ascorbic acid, did not substantially limit the permeability towards hydrogen peroxide. Moreover, different biosensor designs, loading a mixture containing all the components (alone or in combination) and the enzyme, were tested in order to evaluate the changes of the apparent enzyme kinetic parameters, such as V_MAX_ and K_M_, and analytical response in terms of Linear Region Slope, highlighting how the presence of all components (starch, glycerol, and polyethyleneimine) were able to substantially enhance the performance of the biosensors. The surface analysis of the biosensors was performed by scanning electron microscope (SEM). More, it was shown that the same performances were kept unchanged for seven days, proving the suitability of this biosensor design for short- and mid-term use.

## 1. Introduction

In recent years, biosensors have attracted much attention, not only from researchers, but also from companies, because of their wide range of applications. Biosensors are used not only for biomedical purposes, as in analytical devices for the detection of biomarkers for healthcare monitoring or screening for disease, but also for nutraceuticals monitoring in food or food processes, for environmental control, and for the marine sector, as well as for wearable devices’ applications [1]. The widespread use of biosensors is mainly linked to their low cost, simplicity of use, and, above all, their sensitivity and stability, due to the presence of the enzyme acting as a biocatalyst.

One of the most common types of biosensors is the amperometric one, because of its high sensitivity and ease of usage, but also for the possibility of integrating it in miniaturized devices [2]. This kind of biosensor exploits the capability of measuring the concentration of an analyte by means of the electrical signal obtained by the oxidation or reduction of a byproduct, which is generated by an enzymatic reaction, following the highly selective recognition of the molecule under study by the enzyme that is loaded onto the biosensor [3,4].

In the present study, a glucose biosensor was used in order to define the impact of the presence of some enzyme stabilizers on the performance of the biosensor itself.

This biosensor, exploited in the present study, uses the ability of glucose oxidase (GOx) to selectively recognize and convert D-glucose as follows:
β-D-Glucose + FAD^+^-GOx  ⟶ D-Glucono-δ-Lactone + FADH_2_-GOx(1)
FADH_2_-GOx + O_2_ ⟶ FAD^+^-GOx + H_2_O_2_(2)

H_2_O_2_, which is related to the glucose concentrations [3,5,6,7,8,9], is directly oxidized on a platinum surface, when an anodic potential of +700 mV vs. Ag/AgCl is applied, as follows [10]:
H_2_O_2_ ⟶ O_2_ + 2H^+^ + 2e^−^(3)

Given the increasing number of fields of application of biosensors, ample attention has been given to improvement and conservation of biocatalytic functions of enzyme in terms of activity and stability [1,3].

The biosensor stability was extensively investigated in the past [11]. In fact, the author claimed in the paper that the immobilization procedures are fundamental to define the stability of the biosensor, as they affect the preservation of the catalytic activity of the enzyme molecules, and therefore the functioning period and the reusability of the biosensor.

In particular, the immobilization procedures can govern the denaturation process against the enzyme molecules [11], which determines the unfolding of the enzyme tertiary structure, thus affecting the functioning of the enzyme [12]. Unfolding is also responsible for an irreversible loss of activity or inactivation of the enzyme, influencing its conformational (or thermodynamic) stability but also its kinetic (or long-term) stability [12]. Also, noncovalent aggregation has to be taken into account among denaturation processes [13].

As previously published [3], some parameters that can demarcate a change in the immobilized-enzyme stability and specificity must be taken into account in order to evaluate the variations in biosensor performances. In the present study, V_MAX_ and K_M_ have been considered as the main parameters describing the enzymatic kinetic features, and the Linear Region Slope (LRS) as the most important analytical parameter describing the biosensor efficiency [3]. V_MAX_ occurs when the enzyme molecules are entirely saturated by the substrate and represents the highest conversion velocity of the substrate into byproducts [3,14], and has been considered as the main parameter which reflects the number of active molecules loaded onto the biosensor surface [3,6,15,16], while K_M_, the apparent Michaelis–Menten constant, which delineates the substrate concentration, gives half of the V_MAX_, and is linked to enzyme affinity for the substrate [6,17], has been considered an index of the variations of substrate/enzyme binding [3,16], also able to affect the linear operation range; in fact, a higher value of K_M_ produces a wide operational linear range [17,18,19]. Moreover, LRS has been studied since, defining the sensitivity of the biosensor towards the substrate, it results in being the most significant analytical parameter [3,19,20].

Several strategies have been implemented to improve the biosensor performance. One of these concerns the use of molecules that act as enzymatic boosters, as in the case of the polyethyleneimine (PEI). It has been widely shown that PEI is able to influence the enzyme activity by an electrostatic stabilization producing a considerable improvement in biosensor performance [16,17,21,22,23].

More recently, consideration has been given to the impact of using some additives, such as glycerol (GLY), in improving biosensor activity [24]. In fact, it has been largely proven that sugars and GLY can behave as enzyme stabilizers as a consequence of the interactivity with protein molecules, able to avoid the fast inactivation of the enzyme molecules by controlling their microenvironment [24,25].

Moreover, in the literature, it has been widely highlighted how sugars can be employed to conserve enzyme activity. It has been already demonstrated that sugars, as well as glycerol, can endorse the consolidation of hydrophobic interactions among nonpolar amino acids, leading to protein thermostability [26,27], but also that the effect of sugars against protein destabilization is due to the replacement of water molecules removed from the hydration shells of the proteins [26].

In the present study, the impact of the presence of PEI, as enzyme booster, but also of sugars and GLY, as enzyme stabilizers, has been studied on the glucose biosensor parameters V_MAX_, K_M_, and LRS.

## 2. Material and Methods

### 2.1. Chemicals and Solutions

All substances were purchased from Sigma-Aldrich (Milano, Italy). PBS 100 mM (pH = 7.4) was prepared by mixing NaH_2_PO_4_ (6.89 g), NaCl (8.9 g), and NaOH (1.76 g) in 1 L of distilled water. The glucose oxidase from *Aspergillus niger* (GOx, EC 1.1.3.4; Cat: G7141) was used to obtain a final concentration of 895 U/mL. Glucose solution for calibrations (GLU, 1 M) was obtained by solubilizing the powder in distilled water and left for 24 h at room temperature to permit the equilibration of anomers, and then left at 4 °C. Ascorbic acid stock solution (AA, 100 mM) was prepared by dissolving L-ascorbic acid powder in 0.01 M HCl. Hydrogen peroxide solution (HP, 100 mM) was obtained by diluting HP 30% stock solution in ultrapure water. The enzyme stabilizer polyethyleneimine (PEI) solution 1% *w*/*v* was obtained by dilution of the stock solution (50% *w*/*v*; Cat: 18,197-8) in bidistilled water. In the same manner, GLY 1% was prepared by dissolving stock solution (87%) in bidistilled water. The ortho-Phenylenediamine monomer solution (oPD, 300 mM) was made by dissolving the powder in 20 mL of fresh PBS. Starch solutions (GEL 10-5-1-0.5%) were obtained by dissolving starch powder in hot water (100 °C) and stirring them until complete dissolution. Teflon-coated platinum (Pt/Ir, 90% Pt, 10% Ir; ∅ = 125 μm) was acquired from Advent Research Materials (Eynsham, UK).

### 2.2. Instrumentation and Software

All amperometric procedures were carried out by using a conventional three-electrode cell comprising a beaker containing 20 mL of fresh PBS, four glucose biosensors as working electrodes, an Ag/AgCl (3 M) electrode (Bioanalytical Systems, Inc. West Lafayette, IN, USA), and a big-surface stainless steel needle as auxiliary electrode. A four-channel potentiostat (eDAQ Quadstat, e-Corder 410, eDAQ Europe, Poland) was utilized for all electrochemical measures.

### 2.3. Starch-Based Hydrogel Microsensors and Glucose Biosensors Construction and Calibration

Sensors (Figure 1, Panel A) were constructed as previously described [3,9,10,18,24,28,29] by cutting a portion of a Pt/Ir wire and by exposing 2 mm of bare metal, cutting away the Teflon™ insulation, to allow the welding of the wire to a pin, necessary for the connection of the microsensors to the holder. Then, 1 mm of the Teflon™ insulation was cut away at the other end of the wire to obtain the microsensor and biosensor active surface. Initially, to typify starch-based hydrogel layer permeability to HP and AA, a first sequence of experiments was done. Firstly, AA and HP calibrations were done on bare metal by exposing microsensors to 0–1 mM AA concentrations and 0–100 µM HP concentrations, by adding known volumes of AA or HP stock solution (100 mM), respectively. In both cases, an anodic potential of +700 mV vs. Ag/AgCl (NaCl 3M) was applied. Then, the starch-based hydrogel was obtained as follows: dependent on the final concentration of the hydrogel, the starch powder was weighted and dissolved in 2 mL of fresh PBS. The solution was kept under agitation at maximum speed, in a water bath until the water reached 100 °C. The obtained hydrogel was then removed from the heat source and left under stirring until cooling. Then, five starch-based hydrogel layers were deposited and allowed to dry for five minutes, after each dip, at room temperature. After 30 min from the last starch-based hydrogel layering, the electrodeposition of PPD polymer was carried out by applying a constant anodic potential of +700 mV vs. Ag/AgCl for 15 min, when the microsensors were immersed in the monomer solution; polymer is indispensable to block the currents derived from ascorbic acid, which represents the prototype of the interfering species, but also to attempt to increase further microsensor and biosensor selectivity [9,10,30]. The microsensors so manufactured were polarized and left overnight in PBS and calibrated the following day (Day 1) by exposing them to the same HP and AA concentrations as on bare metal. Percentage measurements of 1 mM AA (AA I_lim_) and HP slope versus bare metal values (0% concentration) were evaluated. The protocol was applied using different concentrations of starch-based hydrogel ranging from 0.5% up to 10%.

Based on the same manufacture protocol, different biosensors’ designs were prepared (Figure 1, Panel B). A first experiment was conducted to evaluate the impact of the starch-based hydrogel on the biosensor performances. A basic biosensor design was built, as previously published [3,9,10,18,19,24]. In brief: at Day 0, the 1 mm Pt/Ir active surface was modified by the deposition of the enzyme by means of five dip evaporations, waiting five minutes between each dip, and then the electropolymerization of PPD was carried out as previously mentioned. A second design was made by loading a mixture containing the starch-based hydrogel (5%) and GOx in the final concentration of 895 U/mL. The mixture was loaded five times, waiting five minutes between each immersion. Then, the deposition of PPD was performed. In both cases, after manufacture, biosensors were washed in fresh distilled water and placed in 20 mL of fresh PBS and then polarized at +700 mV vs. Ag/AgCl and left overnight. The following day (Day 1), biosensors were calibrated by exposing them to increasing concentration of AA ranging from 0 up to 1 mM to evaluate the shielding capability against AA. Next, a full calibration was done by adding known volumes of a glucose 1 M solution in a range of concentrations between 0 and 180 mM. Biosensor functioning was evaluated from calibration data in terms of enzymatic kinetic (V_MAX_ and K_M_), analytical performances (LRS).

Starting from these initial results and based on the same building protocol, different biosensors’ designs were manufactured (Figure 1, Panel B), all based on the loading of a mixture containing the starch-based hydrogel (5%), the GOx 895 U/mL, and adding other different components as PEI (1%) and GLY (1%), individually or in combination with each other. In each design, the mixture was loaded by means of five dips, at five-minute intervals from each other. Afterwards, the electrodeposition of PPD was performed. The first design (B1) was built with the deposition of the mixture containing the starch-based hydrogel and the GOx and the deposition of PPD polymer. A second design (B2) was obtained by the deposition of the previously mentioned mixture added with PEI (1%), while the third design (B3) was obtained by loading the mixture added with GLY (1%). The last design (B4) was constructed by loading the mixture added with all the components together. Following the manufacture, biosensors were immersed in 20 mL of fresh PBS and left under a constant potential of +700 mV vs. Ag/AgCl, in order to let the currents stabilize overnight.

All biosensors’ designs were then calibrated at Day 1 in a 20 mL PBS electrochemical cell, exposing them to 0–180 mM of glucose, at room temperature (25 ± 2 °C). Michaelis−Menten enzyme kinetics equation for a two-substrate system was used to fit current data [19] in order to evaluate the V_MAX_ and K_M_ parameters, while a linear regression of data obtained in the 0–2 mM range of glucose concentrations was done in order to extrapolate the LRS value.

At Day 1, AA shielding of biosensors was also evaluated (data not shown) in fresh PBS at room temperature. A fixed potential was used and, after having reached a stabile baseline, biosensors were exposed to 0–1 mM AA concentrations by adding known volumes of AA solution (100 mM).

The aging was assessed on the highest performing biosensor design by repeating the above-described calibration protocol on Day 7, 14, and 21. During the storage period, the biosensors were kept at +4 °C located inside a 25 mL sealed modified Falcon^®®^ test tube, in order to prevent the tips of the biosensors from touching the walls, in dry conditions.

### 2.4. Statistical Analysis

Currents were expressed in nanoampere (nA) and given as mean ± standard error of the mean (nA ± SEM), while I_lim_ parameter, which shows the current value obtained at 1 mM of AA concentration, was chosen as an index of the shielding capability against AA of microsensors and biosensor, as stated previously [31].

Statistical significance (*p* values) was evaluated by means of ANOVA by means of GraphPad Prism 5.02 v software.

### 2.5. Scanning Electron Microscopy

For scanning electron microscopy (SEM) analysis, appropriate sensors were built by modifying the Pt surface, obtained as described in Section 2.3, and by layering the different components individually or in combination. All sensors underwent PPD electropolymerization. After the immobilization on an appropriate holder and the gold coating, microphotographs were obtained using a DSM 962 Zeiss conventional SEM (Oberkochen, Germany) with an accelerating voltage of 25 kV.

## 3. Results and Discussion

### 3.1. Effect of Different Concentrations of Starch in the Hydrogel on the AA Shielding and HP Permeability

As described in Section 2.3, the percentage variations of the calculated HP slopes on bare platinum and on different concentrations of the starch-based hydrogel were evaluated. In Figure 2, Panel A, the deposition of increasing concentrations of the starch-based hydrogel determined a significant (*p* < 0.05 vs. 0%) decrease in HP permeability for all concentrations used, while the 0.5% concentration produced a 20% decrease, and 1% and 5% were around 40% decrease. The maximum diminution of about 75%, if compared with the control, occurred with a starch concentration of 10%

In Panel B, AA percentages are represented. All the starch concentrations in the hydrogel showed a substantial shielding capability against AA, demonstrating that the presence of the hydrogel does not interfere with the electrodeposition of the PPD polymer, since the AA I_lim_ values were in line with those previously published for designs without gel [9,24,30]. All the concentrations produced a significant decrease in AA currents, all over 99%. In particular, the starch at 5% determined a decrease of about 99.7%, while at 10%, of about 99.8%.

Considering the low ability to block the passage of HP and also considering that the deposition of the starch-based hydrogel allows the correct electrodeposition of the PPD polymer, the GEL was found to be an adequate element for the preparation of the biosensors.

More, based on the combined results of HP permeability and AA shielding, the 5% starch-based hydrogel concentration was chosen for the characterization of different biosensor designs.

### 3.2. Effect of Starch-Based Hydrogel (5%) on the Glucose Biosensor Enzymatic and Analytical Performances

In Figure 3, the kinetic and analytical performances of glucose biosensors, depending on the presence or absence of the starch- based hydrogel at 5% concentration, are displayed. As shown in Panel A, the mixture containing the hydrogel and the enzyme produced a significant increase (*p* < 0.05) in V_MAX_ (blue line), which passed from 83.7 ± 4.8 nA in the design loading only the enzyme (red line), to 170.9 ± 10.7 nA in the one loading the mixture. On the contrary, the presence of the mixture determined a decrease in the K_M_, although not significant.

As highlighted in Panel B, even LRS showed a significant increase (*p* < 0.05) in the presence of the mixture hydrogel–enzyme, passing from 4.45 ± 0.04 nA mM^−1^ to 8.29 ± 0.16 nA mM^−1^.

In both groups, AA shielding was in line with those obtained in the presence of only 5% starch-based hydrogel and PPD (data not shown).

From the results, it is possible to highlight that the presence of complex sugars, such as starch, is responsible for a significant increase in the number of active enzyme molecules on the biosensor surface as reflected in the V_MAX_ increase, and consequently that of LRS, as long as the sensitivity to HP remains unchanged in the design [10,20,30], as in this case (data not shown).

The starch-based hydrogel deposition probably is responsible for a better orientation of enzyme molecules, which appear to be arranged in the space in a way that is more favorable to the interaction with the substrate, taking also into account that these molecules are not covalently linked to each other or to the surface of Pt. Moreover, as previously demonstrated [32,33], the presence of the starch could be responsible for the generation of a hydration shell around the proteins, due to the increase in the water surface tension, stabilizing, in fact, the enzyme molecules and enhancing their catalytic activity.

### 3.3. Scanning Electron Microscopy (SEM) Analysis

As previously published [34], a further characterization of biosensors was done by means of SEM, a technique useful in determining the surface features. Microphotographs were taken with a conventional SEM at Day 1 after construction at 5000X magnification (Figure 4).

Surfaces of variously layered sensors (Figure 4B–F) are markedly different from bare Pt (Figure 4A), confirming the deposition of varied layers over the Pt surface. In Panel B of Figure 4B, a lunar-type surface can be observed, which is consistent with a PPD electrogenerated in similar conditions [35]. In Panel C, D, and E, a relatively smooth surface is interrupted by quite rough areas.

It is well acknowledged that heating leads to a complete disruption of starch granules [36]. Consistently with the conditions used for obtaining the gel in the present study, no granular structure comparable in size and shape with those observed from starch can be seen on our SEM imagines. The complete gelification of starch could be responsible for the smoother portion observed in Panel C, D, and E and could be related to gel presence. On the other hand, the granular surface can be due to the PPD layer that emerged from below. The extent of the irregular part seemed to increase when PEI and GLY were added. An even more granular surface is revealed in Panel F of Figure 4, suggesting a synergic effect of both PEI and GLY in determining this extremely lumpy morphology.

### 3.4. Outcomes of the Incorporation of PEI and GLY in the Mixture Containing Starch-Based Hydrogel (5%) and Enzyme

In this research, different biosensor designs were studied, as displayed in Figure 5, in terms of V_MAX_ (Panel A), K_M_ (Panel B), and LRS (Panel C).

As displayed in Panel A, the presence of the other components PEI (1%) and GLY (1%) determined an increase in V_MAX_, resulting in all cases being significantly higher if compared with B1 (*p* < 0.0001). In particular, the presence of PEI (1%) (B2) in the mixture produced an increase of V_MAX_ of about 1.6 times, the presence of GLY (1%) (B3) of about 3.4 times, while the presence of GEL (5%), PEI (1%), and GLY (1%) all together (B4) determined an increase of about 3 times.

On the contrary, K_M_ values (Panel B) showed a substantial and significant decrease when the components (alone or in combination) were added to the mixture (*p* < 0.0001 vs. B1) passing from 16.06 ± 1.35 mM (B1) to ~5.00 mM for all the other designs (B2, B3, B4).

Regarding LRS, this parameter, as displayed in Panel C, underwent a general increase, particularly in B4, reaching the highest value of 73.95 ± 4.81 nA mM^−1^: about nine times higher if compared to the B1 value.

The results of the presence of the studied components in the mixture loaded on the different biosensor designs did not differ from those obtained on 5% starch-based hydrogel and PPD (data not shown).

As highlighted in previous studies [10,37,38], the presence of polycationic compounds such as PEI are able to boost the enzyme activity, a phenomenon also confirmed in the present study by the rise in V_MAX_, and therefore of LRS, most likely related to an increase, although formal, in the number of active molecules present on the surface of the biosensor. The main characteristic of PEI is that of minimizing the electrostatic repulsion produced among the enzyme molecules, globally negatively charged, favoring a better interaction between enzyme and substrate. This phenomenon is also evidenced by the decrease in K_M_. The increase in V_MAX_, and the simultaneous decrease in K_M_, are responsible for an increased LRS, thus determining a better biosensor efficiency.

Moreover, it has been demonstrated [10,39] that GLY can act as an enzyme stabilizer. In the present study, GLY turned out to be more effective in favoring the enzyme activity, so much so that its presence in the biosensor design led to a higher increase in V_MAX_ and LRS, if compared with PEI results. The presence of GLY in the biosensor design favored the GOx activity probably by virtue of its capability of protecting the native structure of the enzyme and favoring its catalytic activity.

The combination of PEI and GLY, combining the effects on electrostatic repulsion and the conservation of the native structure of the protein, led to the highest performing design, in particular, in terms of biosensor efficiency (LRS), derived by a substantial increase of V_MAX_ and a consistent decrease of K_M_. Moreover, the LRS shown by the B4 design, which contained all the components, turned out to be higher when compared with other similar glucose biosensors based on the same geometry [3,9].

### 3.5. Effects of the Presence of PEI and GLY in the Starch-Based Hydrogel (5%) on Glucose Biosensor Stability during Aging

Although B3 and B4 proved to be the highest performing biosensors, the B4 design was chosen for studying the over-time performances. Although it did not show the higher V_MAX_ value, B4 displayed a substantially unchanged K_M_ but, above all, a better LRS, if compared with B3.

As exposed in Figure 6, a period of 21 days was evaluated: each day B4 underwent a full glucose calibration. In Panel A, where variation of V_MAX_ over time is highlighted, it is possible to notice that this parameter suffered a moderate, although not significant, increase passing from 520.0 ± 26.3 nA at Day 1 to 585.9 ± 26.8 nA at Day 7. Subsequently, on Day 14 and Day 21, V_MAX_ suffered a significant, although stable, decrease (*p* < 0.0001 vs. Day 1) resulting in 329.2 ± 27.0 nA and 285.4 ± 12.4 nA, respectively.

As shown in Panel B, K_M_ did not undergo significant changes, showing a slight increase to 4.90 ± 0.52 mM at Day 7, and settling around 3.70 at Day 14 and 21.

Instead, as shown in Panel C, LRS suffered a slight, but not significant, increase at Day 7 (80.23 ± 5.32 nA mM^−1^), if compared with Day 1 (73,95 ± 4,81nA mM^−1^), and presented a significant decrease at Day 14 and 21, when LRS was equal to 47.04 ± 4.59 nA mM^−1^ and 42.04 ± 5.70 nA mM^−1^, respectively.

The aging mechanisms of a biosensor are very complex phenomena and are influenced by each layer and compound used, and they are mainly related to the enzyme activity [6,40], so these kinds of mechanisms are consistently linked to biosensor shelf life. Immobilized enzymes normally undergo denaturation processes, affecting their catalytic capabilities [40].

As previously demonstrated, the presence of polyols and sugars determines an important stabilization on enzyme activity [1,33,41]. In the present study, the simultaneous presence of starch and glycerol produced a two-phase biosensor stability. In addition to creating a substantial improvement in the performance of biosensors on Day 1 (Figure 5), it determined a substantial maintenance of performance up to Day 7, displaying a short, high response stability. Then, from Day 14, a significant decrease occurred in biosensor stability; this turned out to be stable up to Day 21. In the biosensor, the enzyme molecules are not covalently linked to each other, making them prone to rearrangement phenomena. This likely results in an improvement in the interaction between enzyme and substrate molecules, which is evidenced by the increase in V_MAX_ and LRS. All the previously highlighted phenomena could be also likely due to the capability of polysaccharides and polyols of creating a stable hydration shell around the enzyme molecules, which helps in conserving their native conformation (folded state) [1,41], but also establishing some electrostatic interactions with proteins [1], moreover reinforced by the presence of PEI. All these events appear to be extremely important for the protein stabilization and the maintenance of the catalytic activity of the enzyme, also as a result of biosensor storage, which, in the present study, was carried out in the fridge.

The dramatic decrease in biosensor sensitivity during aging could likely be due to the prevalence of deactivation phenomena on the enzyme molecules. The lack of covalent bonds between molecules could be also responsible of the following random or poor orientation of enzymes resulting in partial loss of activity. The decrease in sensitivity monitored in the designs presented in this study was comparable with data found in previous aging experiments obtained from biosensors with the same geometry, when monitored in the same range of time (21 days) and stored at the same temperature (+4 °C) [3].

## 4. Conclusions

In the present study, the role of polysaccharides and polyols, coupled with a polycationic compound, on the glucose biosensor performance has been evaluated. From the results, it is possible to sustain that the mixture containing all the compounds (starch, GLY, and PEI) helps avoid the loss of active enzyme molecules on the transducer surface, being also able to increase their number (at least formally), as highlighted by the increase of V_MAX_ and LRS, but also increase the substrate specificity, making the proposed biosensor design suitable for short- and mid-term use in matrices.

The use of polyols and polysaccharides as enhancers of the enzyme activity needs to be further investigated in order to guarantee the most consistent glucose measurement possible in the matrix.

The proposed biosensors turned out to be higher performing than the previous similar designs [3], in terms of kinetic and analytical parameters, but also in terms of over-time stability. In the presented experiments, we are continuing a study on the use of polyols for the stabilization of immobilized enzymes [24], and because glycols and starch-based hydrogels weren’t used before for glucose biosensors, we are exploring new perspectives in order to ameliorate the biosensor performance for a possible use in different matrices.

At the moment, some experiments are underway to evaluate the role of glycerol and polysaccharides on other biosensors, loading glutamate, and lactate oxidase.

## Figures and Tables

**Figure 1 biosensors-09-00095-f001:**
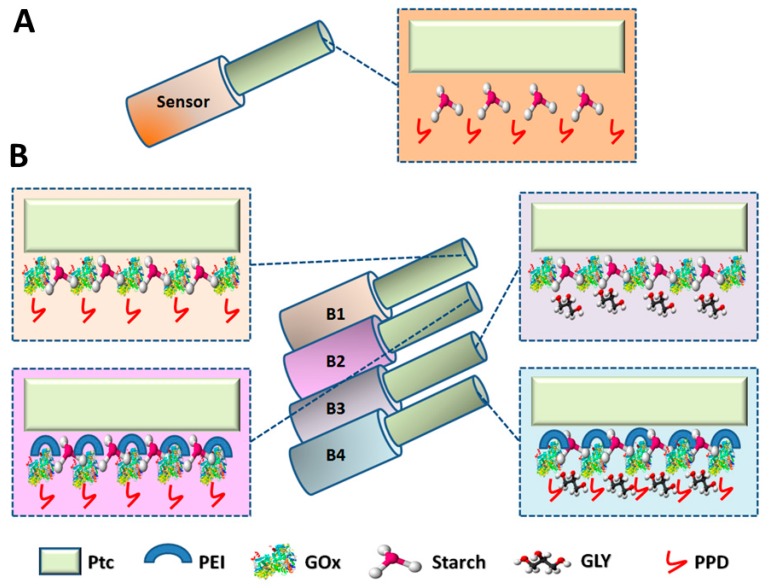
Schematic representation of sensors (Panel **A**) and the main designs of starch-based hydrogel biosensors (Panel **B**) (n = 4) analyzed in this study. B1: Pt_c_/[GEL(5%) + GOx]_5_/PPD; B2: Pt_c_/[GEL(5%) + PEI(1%) + Gox]_5_/PPD; B3: Ptc/[GEL(5%) + GLY(1%) + Gox]_5_/PPD; B4: Pt_c_/[GEL(5%) + PEI(1%) + GLY(1%) + Gox]_5_/PPD. Ptc: Pt cylinder 1 mm long, 125 μm diameter; GEL: starch-based hydrogel; Gox: D-glucose oxidase; PPD: poly-ortho-phenylenediamine; PEI: polyethyleneimine; GLY: glycerol. In brackets: the concentration of the component.

**Figure 2 biosensors-09-00095-f002:**
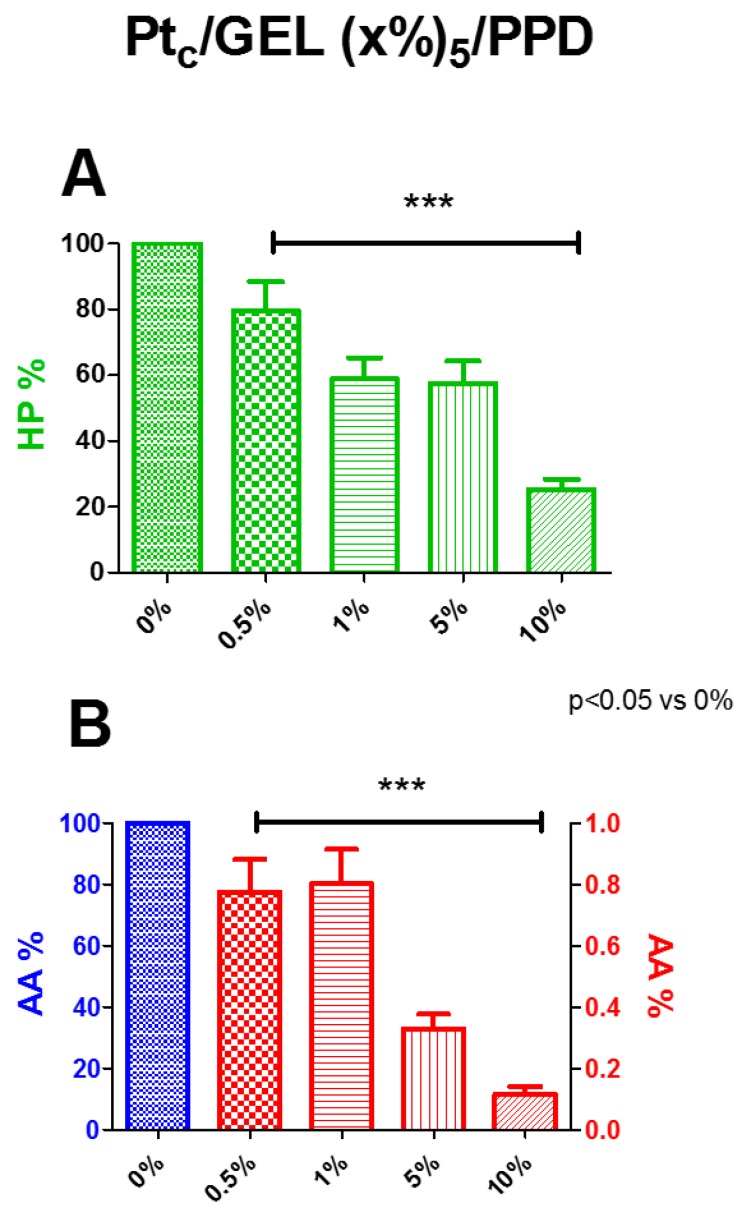
Percentage of HP (Panel **A**) and AA (Panel **B**) measured on microsensor surface depending on the concentration of the starch in the hydrogel. Values are expressed as mean ± SEM. * *p* < 0.05 vs. 0%.

**Figure 3 biosensors-09-00095-f003:**
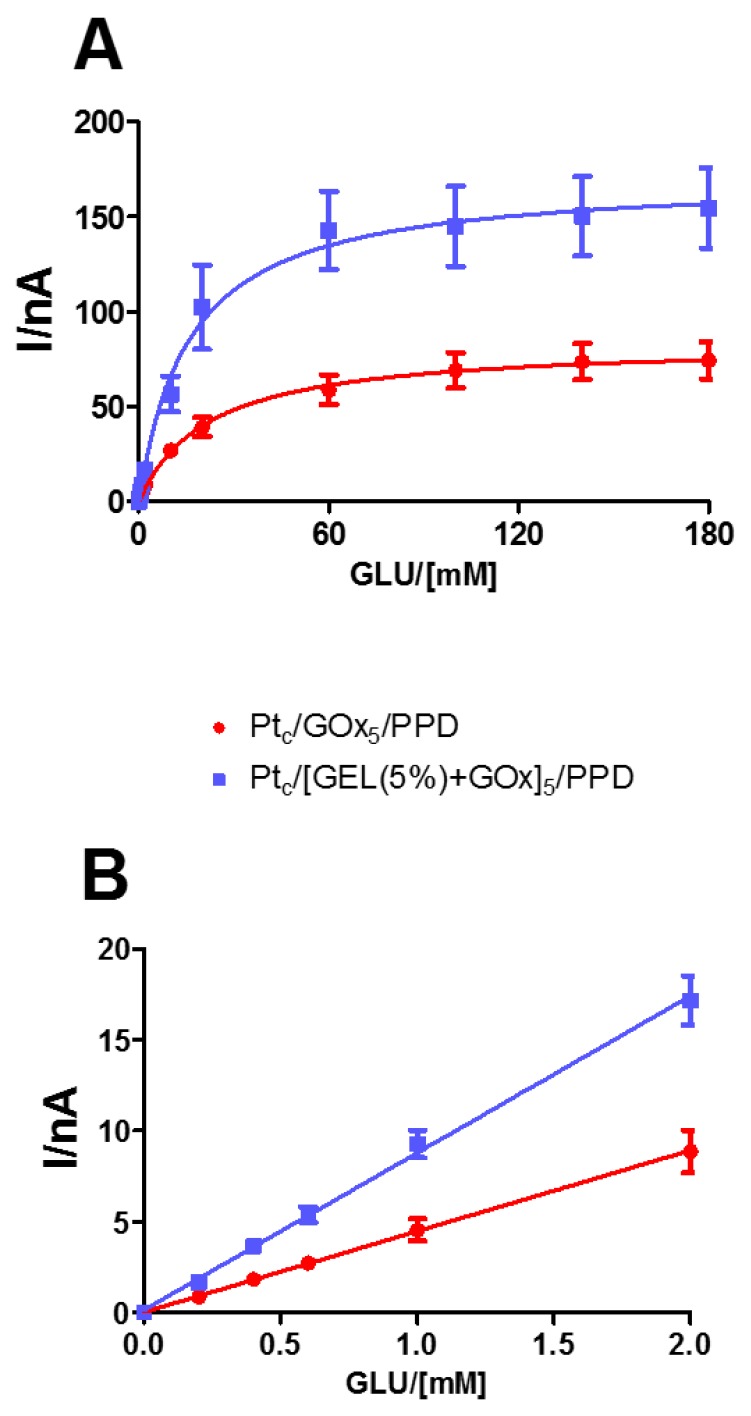
(**A**) Michaelis−Menten kinetics with a range of 0–180 mM and relative 0–2.0 mM linear regression; (**B**) plots of different glucose biosensor designs (n = 4) loading GOx alone (red line: Pt_c_/GOx_5_/PPD) or mixed with starch-based hydrogel (5%) (blue line: Pt_c_/[GEL (5%) + GOx]_5_/PPD). Values are expressed as mean ± SEM. * *p* < 0.05 vs. red line.

**Figure 4 biosensors-09-00095-f004:**
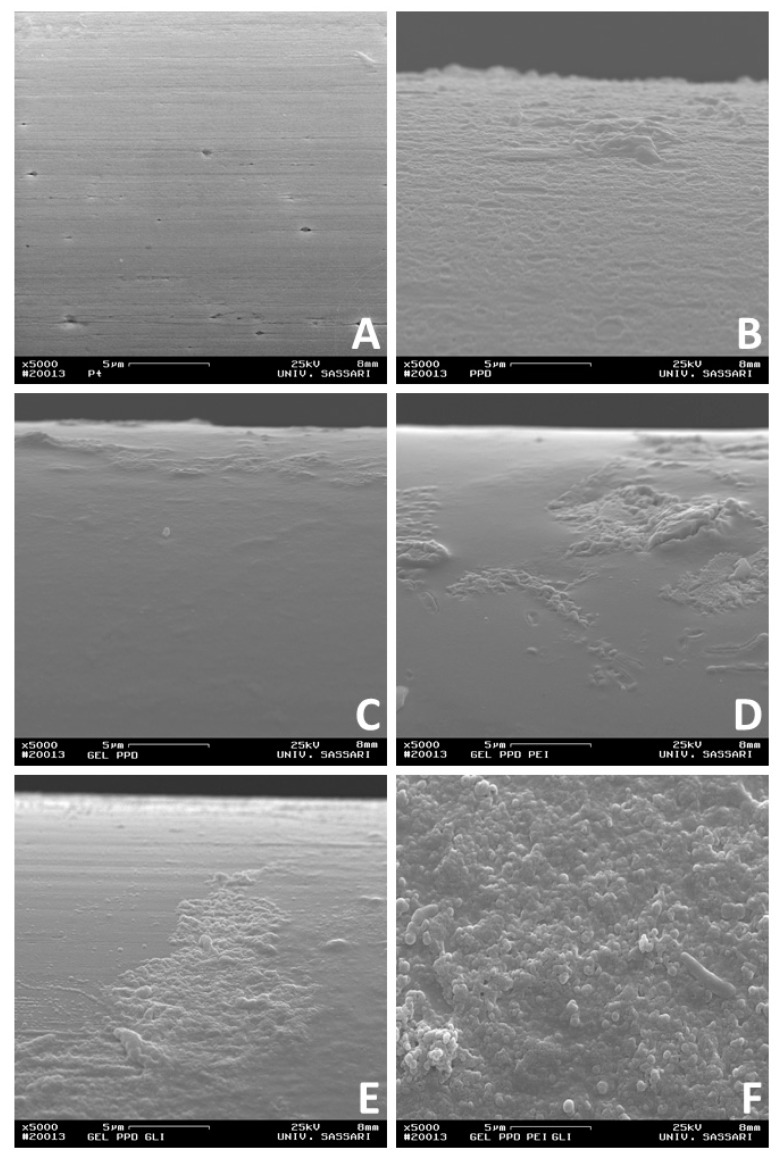
Scanning electron microscope (SEM) of different sensor configurations at 5000X magnification. Ptc (Panel **A**); Ptc/PPD (Panel **B**); Pt_c_/GEL(5%)/PPD (Panel **C**); Pt_c_/[GEL (5%) + PEI (1%)]_5_/PPD (Panel **D**); Pt_c_/[GEL (5%) + GLY (1%)]_5_/PPD (Panel **E**); Pt_c_/GEL (5%) + GLY (1%) + PEI (1%)]_5_/PPD (Panel **F**).

**Figure 5 biosensors-09-00095-f005:**
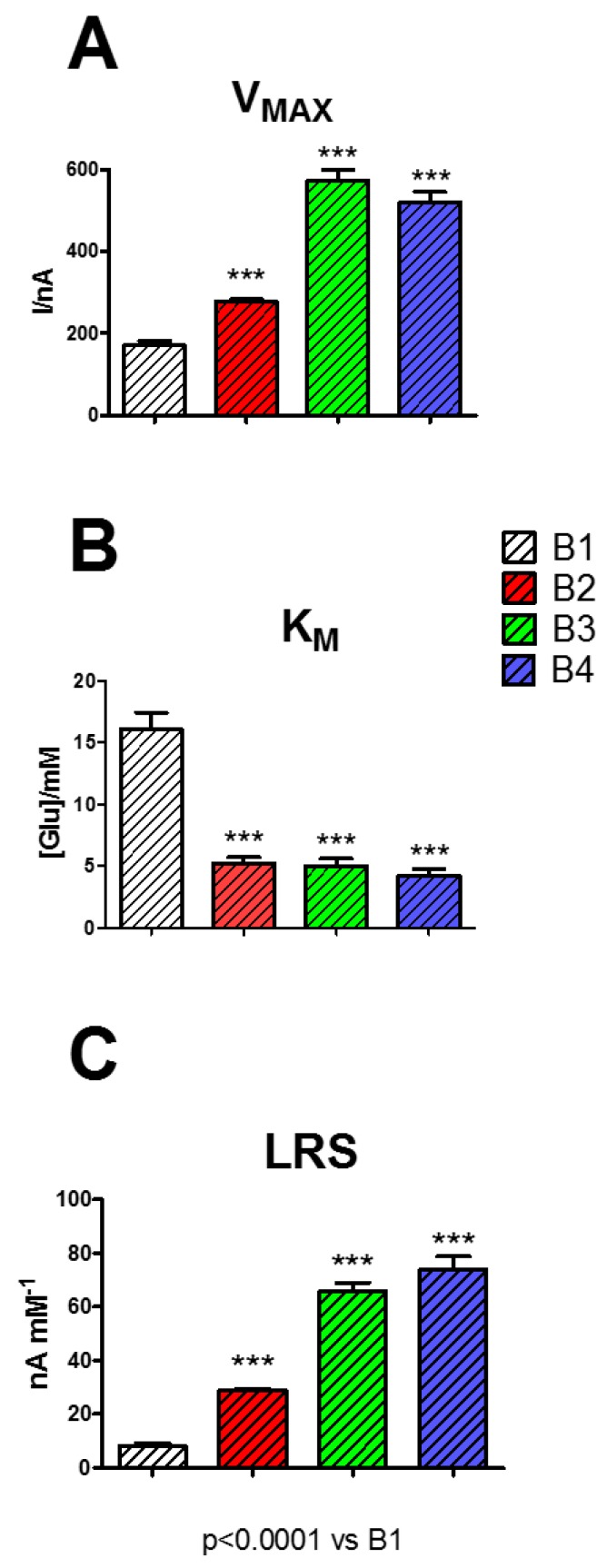
Bar chart displaying the changes of V_MAX_ (Panel **A**), K_M_ (Panel **B**), and LRS (Panel **C**) on different biosensor designs loading five layers of the mixture at different compositions. White bar: B1 [GEL (5%) + GOx]; red bar: B2 [GEL (5%) + PEI (1%) + GOx]; green bar: B3 [GEL (5%) + GLY (1%) + GOx]; blue bar: B4 [GEL (5%) + PEI (1%) + GLY (1%) + GOx]. Values are expressed as mean ± SEM. * *p* < 0.0001 vs. B1.

**Figure 6 biosensors-09-00095-f006:**
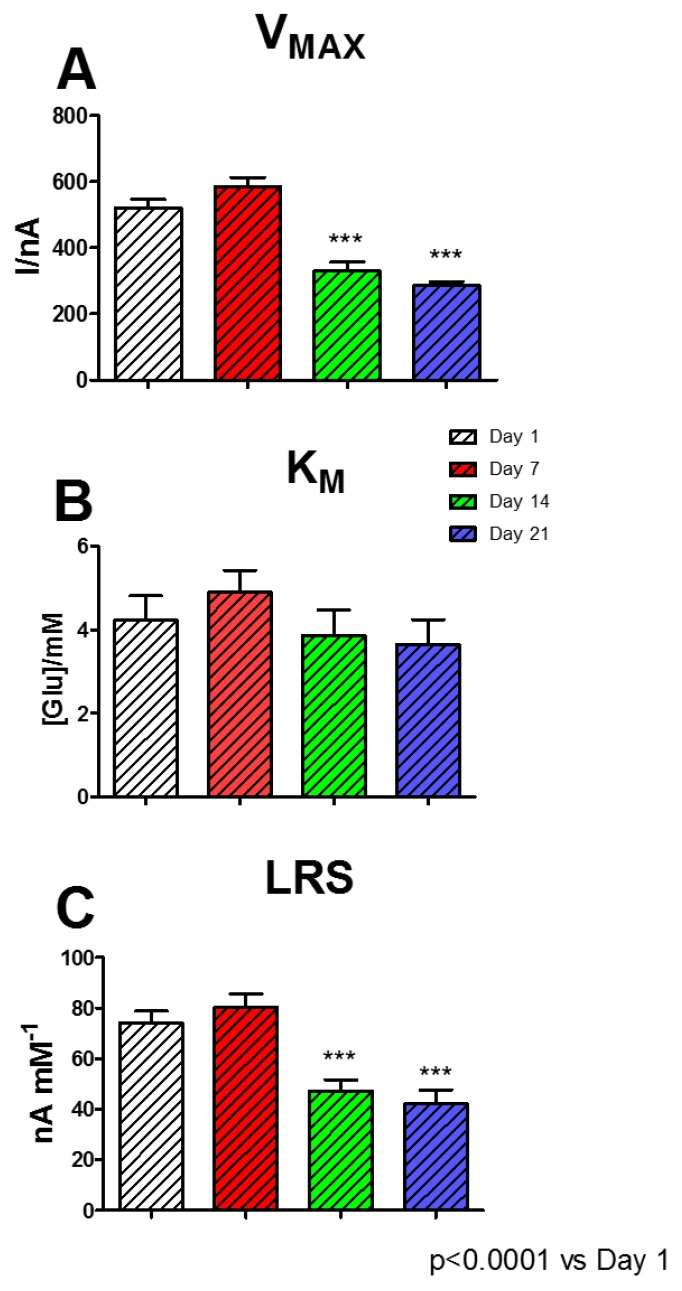
Glucose biosensor stability over time. Bar chart displaying the changes of V_MAX_ (Panel **A**), K_M_ (Panel **B**), and LRS (Panel **C**) on the highest performing biosensor design B4: Pt_c_/[GEL(5%) + PEI(1%) + GLY(1%) + GOx]_5_/PPD, in a period of 21 days. White bar: Day 1; red bar: Day 7; green bar: Day 14; blue bar: Day 21. Values are expressed as mean ± SEM. * *p* < 0.0001 vs. Day 1.

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
