# Peer review of "The Presence of Polysaccharides, Glycerol, and Polyethyleneimine in Hydrogel Enhances the Performance of the Glucose Biosensor"

_biosensors, 2019, doi:10.3390/bios9030095_

Round 1

Reviewer 1 Report

The title is not suitable: think about something related to the conclusion“....role of polysaccharides and polyols, coupled with a polycationic

compound, on the glucose biosensor performances..."

PPD, LRS, PEI … the meaning of each acronyms must be provided only and always in the first time 

Lines

41"One of the most common types of biosensors is the amperometric one." (IMPROVE)

111 all the acronyms were prior mentioned in section 2.1

295 Scanning electron microscopy (SEM) analysis should come before INSTEAD after Optimization

328 Figure 6 is about stability study; 

328 “on the most performing biosensor design” (Must be rewritten)

Author Response

The authors would like to thank the reviewer for the time and attention given to the revision of the manuscript. The observations were punctual, and the quality of the manuscript resulted significantly improved.

Reviewer #1 comments:

The title is not suitable: think about something related to the conclusion“....role of polysaccharides and polyols, coupled with a polycationic compound, on the glucose biosensor performances..."

In acceptance of the reviewer observation, the title has been changed as follows: “The presence of polysaccarides, glycerol and polyethyleneimine in the hydrogel enhances the performance of the glucose biosensor”

PPD, LRS, PEI … the meaning of each acronyms must be provided only and always in the first time

We would like to thank the reviewer for the observation. Acronyms have been provided in the Introduction paragraph and reported along the text

Lines

41"One of the most common types of biosensors is the amperometric one." (IMPROVE)

As requested by the reviewer, the statement about amperometric biosensors has been improved by also adding a reference, which has been added in the reference paragraph as follows. “One of the most common types of biosensors is the amperometric one because of their high sensitivity ease of usage but also for possibility of integrating them in miniaturized [2]. This kind of biosensor exploits the capability of measuring the concentration of an analyte by means of the electrical signal obtained by the oxidation or reduction of a by-product, which is generated by an enzymatic reaction, following the highly selective recognition of the molecule under study by the enzyme that is loaded onto the biosensor [3,4]”. At the same time, depending on this displacement, the numbering of the bibliography was modified in the appropriate paragraph and throughout the text

111 all the acronyms were prior mentioned in section 2.1

As required by the reviewer, where possible, the complete names of the compounds have been replaced by the aforementioned acronyms

295 Scanning electron microscopy (SEM) analysis should come before INSTEAD after Optimization

As suggested by the reviewer, the paragraph “Scanning electron microscopy (SEM) analysis” has been moved before the paragraph “Outcomes of the incorporation of PEI and GLY in the mixture containing starch-based hydrogel (5%) and enzyme”.

328 Figure 6 is about stability study;

In acceptance with the reviewer observation, the caption of the Figure 6 has been improved by adding “Glucose biosensor stability over time.and the title of the paragraph 3.5 has been changed as follows. “Effects of the presence of PEI and GLY in the starch-based hydrogel (5%) on glucose biosensor stability during ageing”.

328 “on the most performing biosensor design” (Must be rewritten)

As noted by the reviewer, the sentence has been reformulated in a more appropriate way, as follows: “The ageing was assessed on the most performing biosensor design by repeating the above-described calibration protocol on Day 7, 14 and 21.”

Reviewer 2 Report

In the paper “A Starch-Based Hydrogel Casted with 3 Polyethyleneimine and Glycerol Enhances Glucose 4 Biosensor Performances” the authors presented the data obtained using different biosensors designs. The presence of glycerol and PEI enhanced the performance of the constructed biosensors. In the paper, the experiments performed are not clearly explained (with many missing details) and there is an important lack of discussion on the obtained results.

General Comments

-Please detail the advantages and disadvantages of the proposed system to others previously developed to obtain biosensors.

-Analytical parameters of the proposed biosensors (linear range and sensitivity) should be compared with those of other authors described in bibliography

-Results of ageing of the proposed biosensors should be compared with those of other authors described in bibliography

-Was the sensor tested using real samples? I really encourage the authors to quantify the glucose content of a real sample using the proposed biosensor and compare the obtained result with HPLC. It is the only way to assure an accurate quantification.

-An example of the obtained raw data during the calibration process of the biosensors an amperometry (current vs time at different glucose additions) should be presented as a supplementary material.

Introduction

Line 75-78. VMAX occurs when the enzyme molecules are entirely saturated by the substrate and represents the highest conversion velocity of the substrate into by-products [2, 13], has been considered as the main parameter which reflects the number of active molecules loaded onto the biosensor surface.

I agree with the authors that Vmax can be used as a rough indicator of the number of active molecules loaded onto the biosensor surface. However, QCM-D analysis is preferred for determining the enzyme conformational changes after the immobilization process (see Colloids and Surfaces B: Biointerfaces 175 (2019) 1–9, https://doi.org/10.1016/j.colsurfb.2018.11.076).  This information should be included in the text.

Section 2. Material and Methods

-Catalog Number from Sigma Aldrich of the used Glucose Oxidase from Aspergillus niger should be included in the manuscript. Since there are many types of GOx available in the market the catalog number must be included. 

-Catalog Number from Sigma Aldrich of the used polyethyleneimine (PEI) should be included in the manuscript.

Section 3. Results and Discussion

-Which is the thickness of the electropolymerized PPD layer? And the thickness of the starch layer? It should be included in the manuscript

-Which is the glucose linear range of sensor B4? It should be included in the manuscript

-Figure 2 should be clearly explained. How the experiments were performed? What was measured?

  -Panel A: How HP% is calculated? Were all experiments performed in a constant HP concentration and the measured current at each sensor configuration compared? If so, which was the HP concentration?

    -Panel B: How AA% is calculated? Why there is a secondary axis? What is the red axis? Why the maximum value in the red axis is 1? Is this a %? Which is the difference with the blue axis?

-Figure 3, Panel B. Y axis is not properly labeled. If this is a calibration curve Y axis should be nA,  not nA·mM-1 (it is the slope of the linear regression).

-Ssection 3.5. There is an important lack of discussion in this section. Results presented in Figure 6 must be deeply discussed and supported using bibliography.

    -Were the sensor parameters measured beyond 21 days? Why 21 days were chosen?

    -How the sensor was preserved between measurements? Submerged in PBS? In a fridge?

    -It can be seen in Figure 6 that the sensor performance is improved from day 1 to day 7 (Vmax is increased and also the sensitivity of the calibration curve). There is a 15% increase in Vmax and LSR. Why the sensor performance is increased? This discussion must be addressed in the manuscript. In addition, the value of LSR in day 1 must be included in the manuscript (see lines 334-335).

    -Lines 324-326. Subsequently, on Day 14 and Day 21, VMAX suffered a significant, although stable, decrease (p < 0.0001 vs Day 1) resulting 329.2 ± 27.0 nA and 285.4 ± 12.4 nA respectively. Why there is such an important decrease? Why the enzyme is deactivated? This discussion must be addressed in the manuscript.

Author Response

Reviewer #2

The authors would like to thank the reviewer for the time and attention given to the revision of the manuscript. The observations were punctual, and the quality of the manuscript resulted significantly improved.

In the paper “A Starch-Based Hydrogel Casted with 3 Polyethyleneimine and Glycerol Enhances Glucose 4 Biosensor Performances” the authors presented the data obtained using different biosensors designs. The presence of glycerol and PEI enhanced the performance of the constructed biosensors. In the paper, the experiments performed are not clearly explained (with many missing details) and there is an important lack of discussion on the obtained results.

General Comments

-Please detail the advantages and disadvantages of the proposed system to others previously developed to obtain biosensors.

We show appreciation for the reviewers’ observation and we would like to highlight that the conclusion paragraph has been amended as follows: “The proposed biosensors design turned out to be more performing than the previous similar design [3], in terms of kinetic and analytical parameters, but also in terms of over-time stability. In the presented experiments we are continuing a study on the use of polyols for the stabilization of immobilized enzymes [24], and because glycols and starch-based hydrogel weren’t used before for glucose biosensors, we are exploring new perspectives in order to ameliorate the biosensor performances for a possible use in different matrices.”

At the moment, some experiments are underway to evaluate the role of glycerol and polysaccharides on other biosensors, loading glutamate and lactate oxidase.”

Authors would also point out that at the moment, the designs presented in this study do not have the useful dimensions for a possible in vivo implantation, but it will be our care to improve the design for a possible implantation in animal models.

-Analytical parameters of the proposed biosensors (linear range and sensitivity) should be compared with those of other authors described in bibliography

We would like to thank the reviewer for the comments. However, because all the parameters, like aging, depend on the type of biosensor design used, which, in turn, depends on the size of the transducer, the type of compounds used, the immobilization method, the enzyme batch, and from many other factors, therefore it is not always possible to report the biosensors data in terms of comparison with others.

Anyway, in acceptance of the reviewer observation, the paragraph “3.4. Outcomes of the incorporation of PEI and GLY in the mixture containing starch-based hydrogel (5%) and enzyme” was amended as follows: “Moreover, LRS showed by B4 design, which contained all the components, turned out to be higher when compared with other similar glucose biosensors based on the same geometry [3, 9].”

-Results of ageing of the proposed biosensors should be compared with those of other authors described in bibliography

We would like to thank the reviewer for the comments. However, because all the parameters, like aging, depend on the type of biosensor design used, which, in turn, depends on the size of the transducer, the type of compounds used, the immobilization method, the enzyme batch, and from many other factors, therefore it is not always possible to report the biosensors data in terms of comparison with others.

Anyway, in acceptance of the reviewer remark, the paragraph "3.5. Effects of the presence of PEI and GLY in the starch-based hydrogel (5%) on glucose biosensor stability during aging.” was amended as follows: ”The decrease in sensitivity monitored in the designs presented in this study were comparable with data found in previous aging experiments obtained from biosensors with the same geometry, when monitored in the same range of time (21 days) and at the same temperature (+4°C) [3].”

-Was the sensor tested using real samples? I really encourage the authors to quantify the glucose content of a real sample using the proposed biosensor and compare the obtained result with HPLC. It is the only way to assure an accurate quantification.

We appreciate the reviewers’ attention on this important point. The proposed biosensors have not been still used with real samples, because this study has been designed for a potential use with biological fluids or with agrifood samples. In this research we concentrated out attention in the characterization of a new design based on a precise mix of components within a hydrogel. The issue of adaptation to different matrices is quite complex (see Rocchitta et al Sensors (Switzerland) 2016, 16.) especially due to the potential interferences, and goes beyond the objectives of this study. This topic will be addressed as a next step of this research.

-An example of the obtained raw data during the calibration process of the biosensors an amperometry (current vs time at different glucose additions) should be presented as a supplementary material.

A representative calibration recording has been showed in the supplementary material as suggested by the reviewer.

Introduction

Line 75-78. VMAX occurs when the enzyme molecules are entirely saturated by the substrate and represents the highest conversion velocity of the substrate into by-products [2, 13], has been considered as the main parameter which reflects the number of active molecules loaded onto the biosensor surface.

I agree with the authors that Vmax can be used as a rough indicator of the number of active molecules loaded onto the biosensor surface. However, QCM-D analysis is preferred for determining the enzyme conformational changes after the immobilization process (see Colloids and Surfaces B: Biointerfaces 175 (2019) 1–9, https://doi.org/10.1016/j.colsurfb.2018.11.076).  This information should be included in the text.

We agree with the reviewers’ observation. To know the enzyme conformational changes after immobilization would be very important in order to explain several phenomena that occur during and after immobilization, e.g the increase (but also the decrease) of VMAX and LRS, that are parameters that depend on enzyme activity.

Unfortunately, the authors do not have the technology to implement the QCM-D analysis, or other technologies, in order to know the conformational state of the enzymes loaded on biosensors, therefore they can be limited to providing observational data based on experimental evidence comparing the data obtained with those obtained on models already validated.

Section 2. Material and Methods

-Catalog Number from Sigma Aldrich of the used Glucose Oxidase from Aspergillus niger should be included in the manuscript. Since there are many types of GOx available in the market the catalog number must be included.  Cat: G7141

-Catalog Number from Sigma Aldrich of the used polyethyleneimine (PEI) should be included in the manuscript. Cat: 18,197-8

As requested by the reviewer, catalog numbers of Glucose Oxidase and Polyethyleneimine have been added in the Material and Methods paragraph and in particular in the section “2.1. Chemicals and solutions”

Section 3. Results and Discussion

-Which is the thickness of the electropolymerized PPD layer? And the thickness of the starch layer? It should be included in the manuscript

We appreciate the reviewer’s remark. Actually, the thickness of PPD has been widely estimated on literature being about 10-30 nm (Malitesta et al Anal. Chem., 62 (1990), p. 2735; Myler et al Anal. Chim. Acta, 357 (1997), p. 55; Craig and O’Neill Analyst, 128 (2003), p. 905; Killoran and O’Neill Electrochimica Acta 53 (2008) 7303–7312) when a polymerization of 15 minutes is performed. For this reason, usually the thickness of PPD polymer is not expressed in the manuscript, because it is taken for granted that the dimensions are those present in the literature.

On the other hand, though, the authors are unable to exactly measure the hydrogel layer. Moreover, when the starch-based biosensors were prepared, authors were able to macroscopically evaluate, under the microscope, that the gel layer on the Pt edge was approximately of the same dimensions of the outer diameter of Pt-Ir Teflon coated wire (about 175 microns), due to the hydration grade of the starch-based hydrogel. When the biosensors were analyzed by means of SEM, the strong dehydration, carried out for the preparation of the specimen, caused a contraction of the gel layer (as shown in Fig 4 panel B -please refer to the new numeration of figures as requested by the other reviewer)

-Which is the glucose linear range of sensor B4? It should be included in the manuscript

In acceptance of the observation made by the reviewer, since the requested data pertains to all biosensor designs, the following sentence was added in the MATERIALS AND METHODS section, in particular in “2.3. Starch-based hydrogel microsensors and Glucose biosensors construction and calibration”: “Michaelis−Menten enzyme kinetics equation for a two-substrate system was used to fit current data [16] in order to deduce VMAX and KM parameters, while a linear regression of data obtained in the 0-2 mM range of glucose concentrations was done in order to extrapolate LRS value.”

-Figure 2 should be clearly explained. How the experiments were performed? What was measured?

  -Panel A: How HP% is calculated? Were all experiments performed in a constant HP concentration and the measured current at each sensor configuration compared? If so, which was the HP concentration?

    -Panel B: How AA% is calculated? Why there is a secondary axis? What is the red axis? Why the maximum value in the red axis is 1? Is this a %? Which is the difference with the blue axis?

Actually, the protocol through which the data of Figure 2 were obtained (Panel A and B) was widely explained in paragraph 2.3 as follows: “Firstly, AA and HP calibrations were done on bare metal by exposing microsensors to 0-1 mM AA concentrations and 0-100 µM HP concentrations, by adding known volumes of AA or HP stock solution (100 mM) respectively. In both cases, an anodic potential of +700 mV vs Ag/AgCl (NaCl 3M) was applied. Then, the starch-based hydrogel was obtained as follows: dependently on the final concentration of the hydrogel, the starch powder was weighted and dissolved in 2 ml of fresh PBS. The solution was kept, under agitation at maximum speed, in a water bath until the water reached 100 °C. The obtained hydrogel was then removed from the heat source and left under stirring until cooling. Then, 5 starch- based hydrogel layers were deposited and let them dry for 5 minutes, after each dip, at room temperature. After 30 minutes from last starch-based hydrogel layering, the electrodeposition of PPD polymer was carried out by applying a constant anodic potential of + 700 mV vs Ag/AgCl for 15 minutes, when the microsensors were immersed in the monomer solution: polymer is indispensable to block the currents derived from ascorbic acid, which represents the prototype of the interfering species, but also to attempt to increase further microsensor and biosensor selectivity [6][7], [27]. The microsensors so manufactured were polarized and left overnight in PBS and calibrated the following day (Day 1) by exposing them to the same HP and AA concentrations as on bare metal. Percentage measurements of 1 mM AA (AA Ilim) and HP slope versus bare metal values (0% concentration) were evaluated. The protocol was applied using different concentrations of starch-based hydrogel ranging from 0.5% up to 10%.”

About Panel B, the secondary axis (the red one) is necessary in order to show both values of AA % on bare metal and on PPD-modified sensors in the same plot. As widely shown in literature (Kirwan et al Sensors (Basel). 2007 ,7(4): 420–437; Rothwell et al Sensors (Basel). 2010; 10(7): 6439–6462; Wynne and Finnerty Chemosensors 2015, 3(2), 55-69), the deposition of PPD layer is mandatory in order to shield currents derived from AA, in particular when biosensors are used in vivo. Actually, the polymer is able to reject more than 99.99% of AA-derived currents, therefore it would become impossible to be able to plot the data in the absence and in the presence of the polymer and simultaneously compare them. The secondary axis is needed to better understand data and to highlight the difference between bare metal (100 %) and polymer-modified sensors (ranging from 0.8 to 0.2 %).

Moreover, the different colors would be used to relate the columns with the respective values (blue column - blue axis, red columns - red axis). More, the red axis has 1% as maximum because, as the polymer determine a very high rejection against AA, the monitored percentage of AA on polymer-modified sensors is very very low, with a maximum of only 1%.

-Figure 3, Panel B. Y axis is not properly labeled. If this is a calibration curve Y axis should be nA,  not nA·mM-1 (it is the slope of the linear regression).

We appreciate the valuable observation of the reviewer. Actually, the mistyping in Figure 3 was corrected and the figure was changed.

-Ssection 3.5. There is an important lack of discussion in this section. Results presented in Figure 6 must be deeply discussed and supported using bibliography.

As requested by the reviewer, authors have implemented the discussion about Figure 6 results, as reported in the following points.

 -Were the sensor parameters measured beyond 21 days? Why 21 days were chosen?

We would like to thank the reviewer for the valuable remark. During experiments, the authors noticed a statistical decrease in the VMAX and LRS parameters, so they decide to continue the observations only up to day 21. Moreover, in a previous study (Puggioni et al 2019 Sensors 19(2):422) we demonstrated that storing the biosensors at-80°C was an important strategy in order to preserve and prolong the biosensor stability. So, the authors weren’t interested in repeating the observation over day 21. At moment, experiments are in progress in order to study the stability of this biosensor design by means of storing it at -80°C.

    -How the sensor was preserved between measurements? Submerged in PBS? In a fridge?

For a better understanding of storage conditions, the paragraph “2.3. Starch-based hydrogel microsensors and Glucose biosensors construction and calibration” has been modified as follows:

“During the storage period, the biosensors were kept at + 4° C located inside a 25 mL sealed modified Falcon® test tube, in order to prevent the tips of the biosensors from touching the walls, in dry conditions.”

    -It can be seen in Figure 6 that the sensor performance is improved from day 1 to day 7 (Vmax is increased and also the sensitivity of the calibration curve). There is a 15% increase in Vmax and LSR. Why the sensor performance is increased? This discussion must be addressed in the manuscript. In addition, the value of LSR in day 1 must be included in the manuscript (see lines 334-335).

As requested by the reviewer, the LRS value at Day 1 was inserted in the manuscript as follows: “Instead, as shown in panel C, LRS suffered a slight, but no significant, increase at Day 7 (80.23 ± 5.32 nA mM-1) ), if compared with Day 1 (73,95 ± 4,81nA mM-1), and presenting a significant decrease at Day 14 and 21, when LRS was equal to 47.04 ± 4.59 nA mM-1 and 42.04 ± 5.70 nA mM-1 respectively.”

    -Lines 324-326. Subsequently, on Day 14 and Day 21, VMAX suffered a significant, although stable, decrease (p < 0.0001 vs Day 1) resulting 329.2 ± 27.0 nA and 285.4 ± 12.4 nA respectively. Why there is such an important decrease? Why the enzyme is deactivated? This discussion must be addressed in the manuscript.

In accordance with the reviewer remark, the discussion about enzyme dropped sensitivity was lacking, so the paragraph “3.5. Effects of the presence of PEI and GLY in the starch-based hydrogel (5%) on glucose biosensor stability during aging” was amended as follows:” The aging mechanisms of a biosensor are very complex phenomena and are influenced by each layer and compound used, and they are mainly related to the enzyme activity [6, 40], so this kind of mechanisms are consistently linked to biosensor shelf life. Immobilized enzymes normally undergo denaturation processes, affecting their catalytic capabilities [40].

As previously demonstrated, the presence of polyols and sugars determines an important stabilization on enzyme activity [1, 33, 41]. In the present study, the simultaneous presence of starch and glycerol, produced a two-phases biosensor stability. In addition to creating a substantial improvement in the performance of biosensors on Day 1 (FIG 5), it determined also a substantial maintenance of performances up to Day 7, displaying a short high response stability. Then, from Day 14, even a significant decrease occurred in biosensor stability, this turned out to be stable up to Day 21. In the biosensor the enzyme molecules are not covalently linked to each other, making them prone to rearrangement phenomena. This likely results in an improvement in the interaction between enzyme and substrate molecules, which is evidenced by the increase in VMAX and LRS. All the previous highlighted phenomena could be also likely due to the capability of polysaccharides and polyols of creating a stable hydration shell around the enzyme molecules, which helps in conserving their native conformation (folded state) [1, 41] but also establishing some electrostatic interactions with proteins [1], moreover reinforced by the presence of PEI. All these events appear to be extremely important for the protein stabilization and the maintenance of the catalytic activity of the enzyme, also as a result of biosensors’ storage, which, in the present study, was carried out in the fridge.

The dramatic decrease in biosensor sensitivity during aging could likely due to the prevalence of deactivation phenomena on the enzyme molecules. The lack of covalent bonds between molecules could be also responsible of the following random or poor orientation of enzymes resulting in partial loss of activity. The decrease in sensitivity monitored in the designs presented in this study were comparable with data found in previous aging experiments obtained from biosensors with the same geometry, when monitored in the same range of time (21 days) and stored at the same temperature (+4°C) [3].”

Round 2

Reviewer 2 Report

The manuscript has been properly modified in order to address all comments and suggestions of the reviewer.